# Apolipoprotein A-IV-Deficient Mice in 129/SvJ Background Are Susceptible to Obesity and Glucose Intolerance

**DOI:** 10.3390/nu15224840

**Published:** 2023-11-20

**Authors:** Fei Wang, Chih-Wei Ko, Jie Qu, Dong Wu, Qi Zhu, Min Liu, Patrick Tso

**Affiliations:** 1Norton Healthcare, 4910 Chamberlain Lane, Louisville, KY 40202, USA; fwang@bellarmine.edu; 2Chroma Medicine, 201 Brookine Ave, Suite 1101, Boston, MA 02215, USA; ckbio87@gmail.com; 3Medpace Reference Laboratories, LLC., 5365 Medpace Way, Cincinnati, OH 45227, USA; j.qu@medpace.com; 4Department of General Surgery, Qilu Hospital of Shandong University, Jinan 250012, China; wudongsdu@hotmail.com; 5Department of Pathology and Laboratory Medicine, Metabolic Diseases Institute, University of Cincinnati, 2180 E Galbraith Road, Cincinnati, OH 45237, USA; zhuqu@ucmail.uc.edu (Q.Z.); lium@ucmail.uc.edu (M.L.)

**Keywords:** Apolipoprotein A-IV, 129/SvJ mice, food intake, obesity, glucose intolerance

## Abstract

Apolipoprotein A-IV (apoA-IV), synthesized by enterocytes, is potentially involved in regulating lipid absorption and metabolism, food intake, and glucose metabolism. In this study, we backcrossed apoA-IV knockout (apoA-IV^−/−^) mice onto the 129/SvJ background for eight generations. Compared to the wild-type (WT) mice, the 129/SvJ apoA-IV^−/−^ mice gained more weight and exhibited delayed glucose clearance even on the chow diet. During a 16-week high-fat diet (20% by weight of fat) study, apoA-IV^−/−^ mice were more obese than the WT mice, which was associated with their increased food intake as well as reduced energy expenditure and physical activity. In addition, apoA-IV^−/−^ mice developed significant insulin resistance (indicated by HOMA-IR) with severe glucose intolerance even though their insulin levels were drastically higher than the WT mice. In conclusion, we have established a model of apoA-IV^−/−^ mice onto the 129/SvJ background. Unlike in the C57BL/6J strain, apoA-IV^−/−^ 129/SvJ mice become significantly more obese and insulin-resistant than WT mice. Our current investigations of apoA-IV in the 129/SvJ strain and our previous studies in the C57BL/6J strain underline the impact of genetic background on apoA-IV metabolic effects.

## 1. Introduction

Obesity is a consequence of an energy imbalance when caloric intake is more than caloric expenditure. Obesity is an established risk factor for type 2 diabetes, characterized by insulin resistance and pancreatic β-cell dysfunction. Prior to type 2 diabetes, susceptible individuals, while exhibiting insulin resistance, are nonetheless capable of maintaining normal blood glucose levels by increasing insulin secretion. However, over time, the excessive secretion of insulin causes β-cell exhaustion, and gradually, type 2 patients can no longer deal with diabetes but become more and more dependent on the supplement of exogenous insulin [1,2]. Thus, improving tissue sensitivity to insulin and enhancing insulin secretion are two ways to treat type 2 diabetes. 

Apolipoprotein A-IV (apoA-IV) is a protein secreted by the small intestine into the lymph in response to fat absorption [3,4]. Various potential physiological functions of apoA-IV have been suggested, including altering lipid absorption [5,6], chylomicron metabolism [7], and metabolic inflammation [8,9]. The role of apoA-IV in regulating insulin and glucose metabolism has been defined in the C57BL/6J mouse strain. C57BL/6J apoA-IV knockout (apoA-IV^−/−^) mice have impaired glucose tolerance secondary to reduced insulin secretion, suggesting a novel incretin role of apoA-IV [10]. One possible mechanism for the role of apoA-IV in insulin secretion is an amplification of insulin exocytosis through the activation of calcium channels. Interestingly, behaving like insulin, apoA-IV is also involved in hepatic glucose metabolism by reducing gluconeogenesis through transcription factors NR1D1 [11]. However, C57BL/6J apoA-IV^−/−^ mice do not exhibit any difference in body weight, body composition, and food intake on a chow diet or after a chronic high-fat diet (HFD) feeding, compared to the wild-type (WT) mice [10]. 

A major strength of using mouse models is that the strain background can greatly influence the phenotype manifested. As an example, the C57BL/6J mice are more susceptible to diet-induced obesity and glucose intolerance compared to other strains, including the 129/SvJ mice [12,13,14,15]. In addition, the C57BL/6J strain has relatively higher levels of blood glucose and insulin concentrations than the 129/SvJ strain [16]. When the apoA-IV^−/−^ mice were first generated in a mixed background of the C57BL/6J and 129/SvJ strains, two distinct phenotypes were observed after chronic HFD feeding. Mice were either extremely obese as their body weight reached 54 g (named “Buddha”) or as lean as 29 g as the WT animals. Notably, the heavier mice had agouti coat color, while the lean ones were black coated (unpublished data). As mentioned above, our previous studies have demonstrated that the C57BL/6J apoA-IV^−/−^ mice had a similar susceptibility to obesity as the WT mice [10]. To further define the impact of mouse genetic background on the apoA-IV metabolic effects, we backcrossed apoA-IV^−/−^ mice onto the 129/SvJ background for eight generations and characterized the difference in metabolic phenotypes between the 129/SvJ WT and apoA-IV^−/−^ mice. Compared to the WT, apoA-IV^−/−^ mice were more susceptible to obesity due to increased dietary intake and decreased energy expenditure. Despite higher insulin and lower lipids in plasma, apoA-IV^−/−^ mice exhibited more insulin resistance than WT mice, and the condition deteriorated over time with the feeding of a high-fat diet.

## 2. Materials and Methods

### 2.1. Animals

Male apoA-IV^−/−^ mice in a mixed 129/SvJ and C57BL/6J background were originally obtained from J. L. Breslow (The Rockefeller University, New York, NY, USA). The 129/SvJ mice were obtained from The Jackson Laboratory. The 129/SvJ apoA-IV^−/−^ mice were generated by backcrossing the background-mixed apoA-IV^−/−^ mice onto the 129/SvJ mice for 8 generations. The absence of apoA-IV was confirmed by PCR amplification of genomic DNA. All mice were bred and housed in our Association for the Assessment and Accreditation of Laboratory Animal Care (AAALAC)-accredited facility, maintained at 22 °C and under a 12:12 h light–dark cycle with ad libitum access to water and food. The standard chow diet, Teklad 7912, contains 5.8% fat by weight, providing 17% of total calories (3.1 kcal/g) (Teklad Diets, Madison, WI, USA). The HFD, Research Diet D03082706, contains 20% fat by weight, providing 40% of total calories (4.54 kcal/g) (Research Diets, New Brunswick, NJ, USA). Body weight and food intake were monitored weekly throughout the study. All animal protocols were approved by the University of Cincinnati’s Institutional Animal Care and Use Committee (IACUC) and complied with the National Institutes of Health Guide for the Care and Use of Laboratory Animals.

### 2.2. Body Composition

Body composition in age-matched apoA-IV^−/−^ and WT mice was assessed via an EchoMRI-100 whole-body composition analyzer (Echo Medical System, Houston, TX, USA). The EchoMRI is a quantitative nuclear magnetic resonance instrument that measures precisely total body fat, lean mass, body fluids, and total body water in living rodents. The fat (lean) percentage was further calculated by dividing the fat (lean) mass to body weight and multiplying the ratio by 100. 

### 2.3. Lipid Absorption

Individually housed mice will be fed ad-lib a semi-synthetic butter oil diet containing 5% sucrose polybehenate (kindly donated by the Procter and Gamble Company), a non-absorbable food additive. On Days 3 and 4, fecal pellets were collected, saponified, methylated, and analyzed by gas chromatography. Fat absorption was calculated from the difference between diet and fecal ratio of behenate (C 22:0) to other fatty acids [17].

### 2.4. Indirect Calorimetry

Mice were individually housed and acclimated to the indirect calorimetry cages for 1 day before data were collected. Indirect calorimetry was performed using a PhysioScan open-circuit Oxymax system v 5.35, a component of the Comprehensive Laboratory Animal Monitoring System (CLAMS; Columbus Instruments, Columbus, OH, USA) to monitor oxygen (O_2_) and carbon dioxide (CO_2_) gas fractions at the inlet and outlet ports in each chamber. The airflow was 0.5 L/min. Sampled air is then passed through O_2_ and CO_2_ sensors to determine O_2_ and CO_2_ content. Mice were maintained at 25 °C with free access to food and water for 24 h. The respiration quotient (RQ) was calculated by the ratio of CO_2_ produced and O_2_ consumed (VO_2_) by mice. Heat production was calculated from the following equation: (3.82 + 1.23 × RQ) × VO_2_ and was represented as Cal/h/kg BW.

### 2.5. Locomotive Activity 

The activity of individually housed mice was evaluated on a relative, not absolute, basis using an eight-cage rack OPTO-M3 Sensor system (Columbus Instruments, Columbus, OH, USA). Home cages were placed in Smart Frame stainless steel cage rack frames (Hamilton-Kinder, Poway, CA, USA). Infrared photobeam interruption sensors mounted in the frames detected each animal’s movements. Activity counts (beam interruptions) were recorded every 60 min throughout the light and dark cycles.

### 2.6. Glucose and Insulin Tolerance Tests

Intraperitoneal glucose tolerance tests (IPGTT) were performed in mice at 12 weeks old on a chow diet and at Week 4, 8, 12, and 16 of the HFD. The mice were fasted for 5 h (from 09:00 h to 14:00 h) and subsequently i.p. injected with 2 g/kg BW of glucose. Tail blood samples were collected before, at 15, 30, 60, and 120 min after injection. Insulin tolerance tests were performed in mice at Week 13 of the HFD. After a 5 h fast (09:00 h to 14:00 h), mice were injected with 0.75 U/kg of Humulin R (Eli Lilly, Indianapolis, IN, USA). Tail blood samples were collected before, at 15, 30, 45, and 60 min after injection. 

### 2.7. Biochemical Analyses

Blood glucose levels from 5 h fasting mice (G_0_) and mice during tolerance tests were measured by a glucometer (Freestyle Lite; Abbott, Chicago, IL, USA). Plasma insulin levels from 5 h fasting mice (I_0_) and mice during tolerance tests were measured by ELISA according to the manufacturer’s manual (EMD Millipore, Billerica, MA, USA). The homeostatic model assessment of insulin resistance (HOMA-IR) was calculated using the equations [(G_0_ × I_0_)/405)], where G_0_ is in mg/dL and I_0_ is in mU/L. Plasma cholesterol levels were measured by Infinity Total Cholesterol Assay Kit (Fisher Scientific, Waltham, MA, USA). Plasma triglycerides were measured by the Randox Triglyceride Assay Kit (Randox Laboratories, Kearneysville, WV, USA). Plasma phospholipids were assessed by Phospholipids C Assay (Wako Life Sciences, Inc., Mountain View, CA, USA). Plasma non-esterified fatty acids were measured by The Wako HR series NEFA-HR (2) (Wako Life Sciences, Inc., Mountain View, CA, USA). All measurements followed the manufacturer’s instructions. Leptin and adiponectin from white adipose tissue were measured by ELISA according to the manufacturer’s manual (EMD Millipore, Billerica, MA, USA).

### 2.8. Islet Morphology

At the end of 16-week HFD, the pancreata of mice were isolated from 5 h fasted mice (from 09:00 h to 14:00 h), fixed in 4% paraformaldehyde, and embedded in paraffin. Sections were prepared, stained with hematoxylin and eosin (H&E), and visualized under a microscope (Olympus BX61; Nikon, Tokyo, Japan).

### 2.9. Statistics

All data are presented as mean ± standard error of the mean (SEM). Interactions between genotypes (apoA-IV^−/−^ vs. WT) were analyzed by two-way repeated-measures ANOVA when appropriate. One-way ANOVA and independent *t*-tests were also used when appropriate. A *p* value of less than 0.05 was considered statistically significant. Statistical comparisons were performed using PRISM 8.0 software (GraphPad Software, La Jolla, CA, USA). 

## 3. Results

### 3.1. ApoA-IV^−/−^ Mice Are Susceptible to Obesity

Before being fed an HFD, apoA-IV^−/−^ mice exhibited significantly greater body weight than WT mice at 12 weeks on a regular chow diet. At 12 weeks old, WT and apoA-IV^−/−^ mice started the HFD feeding for an additional 16 weeks. During the period, apoA-IV^−/−^ mice gained more weight than the WT, leading to a larger difference in body weight between WT and apoA-IV^−/−^ mice (Figure 1A). At the end of the study, apoA-IV^−/−^ mice gained 16.1 ± 1.6 g, whereas WT mice only gained 9.4 ± 0.7 g (*p* < 0.0001). Body composition was performed in WT and apoA-IV^−/−^ mice over time on the HFD. WT and apoA-IV^−/−^ mice increased fat deposition during the period. However, compared to WT mice, apoA-IV^−/−^ mice retained higher percentages of fat and lower percentages of lean mass throughout the study (Figure 1B). To study the contribution of calorie input to the development of obesity, food intake was also measured weekly and converted to calorie intake by the energy intensity of the diet. Throughout the study, the calories consumed by apoA-IV^−/−^ mice remained higher than WT mice, except for the first week, while apoA-IV^−/−^ mice consumed ~35% more diet than the WT (Figure 1C). To further determine if the difference in calorie intake between WT and apoA-IV^−/−^ mice is related to the efficiency of intestinal lipid uptake, lipid absorption was assessed. Oddly, apoA-IV^−/−^ mice absorbed around 8.6% less lipid than WT mice (Figure 1D). 

Indirect calorimetry was performed at Week 8 and the end of 16-week HFD to assess energy metabolism in WT and apoA-IV^−/−^ mice. As shown in Figure 2, in both Weeks 8 and 16, the metabolic rates presented by oxygen consumption in apoA-IV^−/−^ mice were lower than in WT mice. In addition, with a similar respiratory quotient, energy expenditure in apoA-IV^−/−^ mice was significantly lower than in WT mice. ApoA-IV^−/−^ mice also exhibited a significant decrease in cumulative locomotor activity at the end of a 24 h measurement (Appendix A). Body temperature in apoA-IV^−/−^ mice was also significantly lower than in WT mice (36.3 ± 0.1 vs. 36.9 ± 0.1 °C, *p* < 0.01).

### 3.2. ApoA-IV^−/−^ Mice Are Insulin-Resistant with Elevated Circulating Insulin Levels

Basal plasma glucose and insulin were analyzed in mice after a 5 h fast (from 09:00 h to 14:00 h) before the feeding of HFD and then every 2 weeks on HFD. On chow diet, apoA-IV^−/−^ mice already had significantly higher plasma insulin levels than WT mice despite similar basal glucose levels. On HFD, insulin levels in apoA-IV^−/−^ mice gradually increased until the 10th week and then rose sharply relative to WT mice, which persisted until Week 16 (Figure 3A,B). Notably, after Week 14, when apoA-IV^−/−^ mice secreted 10-fold higher insulin than WT mice, basal glucose levels in apoA-IV^−/−^ mice were significantly higher than WT mice. The impaired insulin levels in apoA-IV^−/−^ mice also affected their HOMA-IR indexes. As shown in Figure 3C, apoA-IV^−/−^ mice already developed significant insulin resistance 12 weeks before HFD (HOMA-IR > 5). Over time, on HFD, the severity of insulin resistance in apoA-IV^−/−^ mice increased, positively correlated to the changes in basal insulin levels. On the other hand, although WT mice also developed insulin resistance after Week 2 on HFD, they did not deteriorate further along with the period. 

As the degree of HOMA-IR in apoA-IV^−/−^ mice reflects basal insulin levels, we further compared the morphology of pancreatic islets in WT and apoA-IV^−/−^ mice at the end of 16-week HFD. Figure 4A presents the islet sections with H&E staining. After quantification, there was no difference in the total number of beta islet between WT and apoA-IV^−/−^ mice; however, the average size of the beta islet in apoA-IV^−/−^ mice were significantly larger than the WT (Figure 4B,C). Specifically, around 42.1% of the islets in apoA-IV^−/−^ mice were larger than 5000 µm, while only 15.1% of WT islets were in this size range.

Elevated insulin secretion is closely associated with the disruptions of glucose metabolism. To determine that, intraperitoneal glucose tolerance tests were performed in WT and apoA-IV^−/−^ mice before and every 4 weeks after HFD feeding. On a regular chow diet, the glucose excursion in apoA-IV^−/−^ mice had a delayed reduction 30 min after glucose administration, compared to WT mice (Figure 5A). During the period, apoA-IV^−/−^ mice secreted significantly more insulin than the WT (Figure 5B). Over time on the HFD, WT and apoA-IV^−/−^ mice became more glucose-intolerant, yet glucose levels in apoA-IV^−/−^ mice were consistently higher than in WT mice (Figure 6A). On the other hand, the rise of insulin levels over the time of HFD was only observed in apoA-IV^−/−^ mice but not in WT mice (Figure 6B). An insulin sensitivity test was also performed on Week 13 of HFD. After insulin administration, apoA-IV^−/−^ mice could not metabolize the glucose as efficiently as WT mice, although the insulin levels in apoA-IV^−/−^ mice were much higher than the WT during the same period (Figure 7A,B). 

Circulating lipid contents closely associated with the degree of insulin resistance were measured in WT and apoA-IV^−/−^ mice on Week 8 and Week 16 after HFD feeding. At both time points, apoA-IV^−/−^ mice featured lower levels of plasma cholesterol contents and phospholipids than WT mice, while there were no differences in their levels of triglycerides and NEFA (Figure 8A,B). The expressions of adiponectin and leptin that represent the biological function of adipose tissue were also measured at the end of HFD. While the adiponectin levels are comparable, the adipose tissue of apoA-IV^−/−^ mice expressed significantly higher leptin than WT mice (Figure 8C).

## 4. Discussion 

Increasing evidence indicates that apoA-IV is very important in controlling weight gain with metabolic regulations of food intake, lipid absorption, and lipid and glucose metabolism [18,19]. Although our previous study has demonstrated that deleting apoA-IV in C57BL/6J mice exhibited comparable susceptibility to obesity as WT mice [10], the current study backcrossing apoA-IV^−/−^ mice onto the 129/SvJ background has characterized several significant metabolic phenotypes that emphasize the effect of apoA-IV on weight control. Compared to the WT mice, the 129/SvJ apoA-IV^−/−^ mice had increased body weight and fat mass even at 12 weeks on the chow diet. When fed HFD, apoA-IV^−/−^ mice were significantly more obese than the WT, which was associated with greater fat deposition. Increased energy storage that leads to obesity may be due to increased energy intake and/or reduced energy expenditure. ApoA-IV is well established to regulate food intake. Fujimoto et al. demonstrated that intravenous infusion of apoA-IV-containing chylous lymph in rodents exerted a significant anorectic effect [20]. Subsequent studies showed that inhibition of food intake by apoA-IV also occurs centrally [19]. In the study that generated the original apoA-IV^−/−^ mice, increased food intake was observed in 129/SvJ apoA-IV^−/−^ mice but not in mice with a mixed background of 129/SvJ and C57BL/6J strains [21]. Consistent with previous observations, our current apoA-IV^−/−^ mice consumed more foods than WT mice on the chow diet, the effect of which was more profound on the HFD. Our previous studies in C57BL/6J mice also showed the lack of apoA-IV action on food intake. Together, our findings in these two backgrounds highlight the importance of genetic background on the anorectic function of apoA-IV. However, the mechanisms by which apoA-IV regulates food intake in 129/SvJ but not C57BL/6J mice require further investigation. While it is complicated to identify the modifying gene/s that interact with apoA-IV in these two different background mice, a thorough understanding of the modifying genes using advanced technology, e.g., RNA sequence, will be invaluable to human biology and the opportunity to develop drugs for human obesity and diabetes.

Although apoA-IV^−/−^ mice consumed more calories from diet, they absorbed less lipid than WT mice. This effect of apoA-IV deficiency, however, was only observed in 129/SvJ but not the C57BL/6J strain. The interplay of dietary lipid consumption and intestinal lipid absorption resulting from apoA-IV deficiency remains to be determined. In the current study, it is important to understand if the overall amount of lipids entering the body is different between WT and apoA-IV^−/−^ mice, which may account for their body weight difference. To determine the amount of lipid input during a 16-week HFD, we applied the following calculation: Lipid input (g) = Diet (g/day) × fat in diet (20% by gram) × intestinal lipid uptake (%) × Duration (112 days). During 16-week HFD, the total amount of lipid intake of WT mice was, on average, 56.1 g, while the apoA-IV^−/−^ mice took up 62.5 g of lipids. The difference in lipid input between WT and apoA-IV^−/−^ mice was 6.4 g. If the rates of lipid metabolism in WT and apoA-IV^−/−^ mice are comparable, the greater uptake and accumulation of lipids in apoA-IV^−/−^ mice can be one of the key factors in the development of obesity. 

Besides the increase in energy intake, obesity can also result from an overall energy expense reduction. Thus, we further studied the degree of energy expenditure and fuel preference in WT and apoA-IV^−/−^ mice. Compared to WT mice, apoA-IV^−/−^ mice exhibited lower metabolic rate and energy expenditure, which was associated with reduced heat production, body temperature, and general activity. This is consistent with our previous report in C57BL/6J apoA-IV^−/−^ mice [22]. No difference in the preference of energy sources was observed between WT and apoA-IV^−/−^ mice, as reflected by their comparable respiratory quotient. Collectively, our findings strongly suggest that apoA-IV in 129/SvJ mice plays an important role in energy balance by enhancing calorie intake and reducing energy expenses. It is reported that an animal’s ingestive behavior and energy expenditure are well determined by the populations of neurons in the central melanocortin system [23,24]. For example, agouti-related peptide (AgRP) and neuropeptide Y (NPY) neurons act to not only increase food intake but also reduce energy expenditure, while pro-opiomelanocortin (POMC) neurons function the opposite [25]. Importantly, previous evidence showed that apoA-IV inhibits AgRP/NPY neurons and activates POMC neurons in the arcuate nucleus [26]. Thus, our findings of reduced energy expenditure in the apoA-IV-deficient animals provided novel evidence supporting the apoA-IV action in the central melanocortin system.

ApoA-IV deficiency in the C57BL/6J mice was glucose-intolerant as their β-islets could not secrete more insulin during the first-phase response to elevated glucose compared to WT mice [10]. Administration of apoA-IV to β-islets increases their insulin secretion in the presence of elevated glucose in a dose-dependent manner [10]. However, in our current study using the 129/SvJ mice, the opposite phenotype was observed in apoA-IV^−/−^ mice. Compared to the WT mice, the insulin level in apoA-IV^−/−^ mice was much higher, even on a chow diet. Over the time of HFD, apoA-IV^−/−^ mice developed more severe insulin resistance than WT mice, as indicated by HOMA-IR values. Importantly, the sudden increase of HOMA-IR in apoA-IV^−/−^ mice occurring at Week 12 was primarily associated with the extreme rise of plasma insulin level. However, starting from Week 14, when the insulin level in apoA-IV^−/−^ mice was even 5-fold higher than in WT mice, apoA-IV^−/−^ mice were unable to maintain basal glucose levels and the WT. Our data suggest that apoA-IV deficiency in the 129/SvJ strain increases the demand for insulin secretion to maintain glycemic control. 

Hyperinsulinemia and insulin resistance are expected to enlarge the pancreatic islet mass, primarily resulting from hypertrophy or increased β-cells. When we examined the pancreas at the end of the 16-week HFD feeding, we found that while the number of pancreatic β-islets was comparable between apoA-IV^−/−^ and WT mice, apoA-IV^−/−^ mice tended to have larger sizes of β-islets than the WT mice. This finding supports the notion that the increased insulin secretion by the pancreas of apoA-IV^−/−^ mice may be due to the compensatory response to overcome the dramatic insulin resistance. To further determine the dysfunction of glucose handling in apoA-IV^−/−^ mice, IPGTT was first assessed. After glucose administration, apoA-IV^−/−^ mice exhibited abnormal glycemic excursion with significantly higher insulin levels. This impaired glucose tolerance became more severe over time on HFD feeding. The insulin tolerance test further informed us of the degree of insulin resistance in mice. It was demonstrated that apoA-IV^−/−^ mice developed more insulin resistance than WT mice during HFD feeding. Thus, apoA-IV is intimately involved in insulin secretion as well as insulin sensitivity. 

Dyslipidemia is an essential component of metabolic syndrome that increases the risk of insulin resistance and type 2 diabetes mellitus. Obesity promotes increased delivery of fatty acids to the liver and enhances the synthesis of very low-density lipoprotein, leading to hypertriglyceridemia [27,28]. Nonetheless, apoA-IV^−/−^ mice had reduced circulating lipids compared to the WT mice, even on a standard chow diet. Further studies are required to measure mRNA and protein levels of essential genes that regulate lipid synthesis and metabolism in 129/SvJ mice.

Circulating NEFAs, on the other hand, have been identified as a critical factor in modulating insulin sensitivity [29,30]. Insulin resistance of the adipocyte is shown as the initiating insult, leading to increased intracellular hydrolysis of triglycerides and release of NEFAs into the circulation [31,32]. In humans, insulin resistance develops within hours of an acute increase in plasma NEFA levels [33]. In the current study, when apoA-IV^−/−^ mice developed significant insulin resistance after 16-week HFD, their circulating NEFA levels were comparable to WT mice. Together, our findings strongly imply that apoA-IV in the 129/SvJ mice plays a very important role in regulating tissue sensitivity to insulin, the effect of which did not appear to be dependent on dyslipidemia. Besides dyslipidemia, adipose tissue also has a major role in the pathogenesis of insulin resistance, which is associated with the secretion of adipokines, especially adiponectin and leptin. The expression of adiponectin and its plasma level are inversely associated with insulin resistance [34,35]. In the current study, although there was a dramatic difference in insulin resistance levels between WT and apoA-IV^−/−^ mice, apoA-IV^−/−^ mice had comparable expression of adipose tissue adiponectin to the WT. However, the expression of tissue leptin was significantly higher in apoA-IV^−/−^ mice than in WT mice. Further studies are required to determine how apoA-IV modulates insulin sensitivity in 129/SvJ mice. 

## 5. Conclusions

In conclusion, unlike the phenotypes observed in the C57BL/6J mice, apoA-IV in the 129/SvJ mice has remarked regulations in energy balance (food intake and energy expenditure) and glucose homeostasis (glucose tolerance and insulin sensitivities). The fact that apoA-IV deficiency dramatically affects the 129/SvJ, but not the C57BL/6J strain, is important and implies other gene(s)/factor(s) that modify apoA-IV’s actions in these two strains. Similar phenomena probably exist in humans since genetic polymorphisms in human apoA-IV have been associated with differential effects on lipid metabolism, adiposity, and body mass [36]. Our current investigations of apoA-IV in the 129/SvJ strain and our previous studies in the C57BL/6J strain highlight the impact of genetic background on apoA-IV metabolic effects. We understand that these apoA-IV^−/−^ mice are inbred and human populations are genetically diverse. Recently developed outbred mouse populations, such as the diversity outbred (DO) mice, available at The Jackson Laboratory, have created research options that parallel or even exceed human genetic diversity [37]. Therefore, the effects of apoA-IV deficiency on energy and glucose metabolisms in the DO mice should be further pursued, and the findings from the inbred and outbred apoA-IV^−/−^ mice will stimulate future research in individualized treatments of obesity and diabetes in humans in the era of precision medicine.

## Figures and Tables

**Figure 1 nutrients-15-04840-f001:**
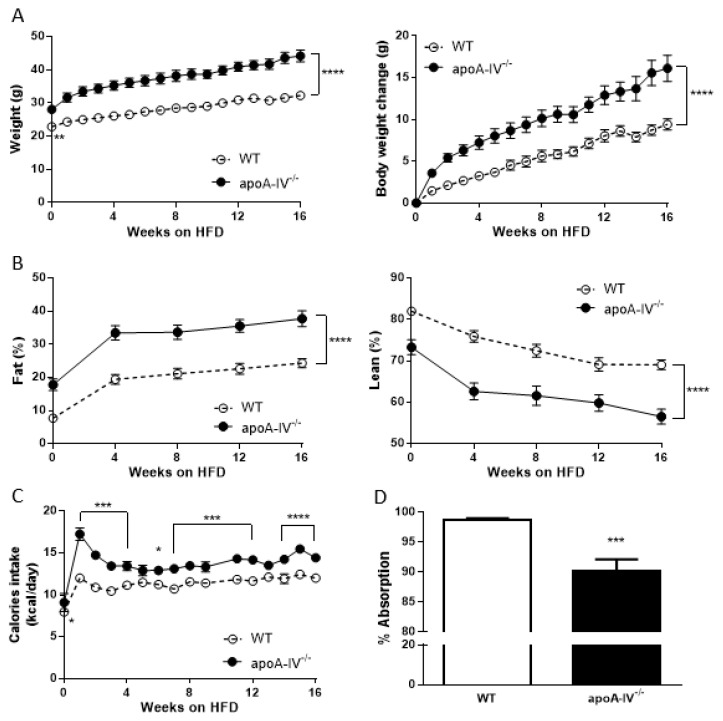
Growth curve during 16-week HFD. Comparison of body weight (**A**), body composition (**B**), calorie intake (**C**), and lipid absorption (**D**) between the apoA-IV^−/−^ and the WT mice. Values are expressed as means ± SEM. *n* = 11 in WT group, *n* = 7 in apoA-IV^−/−^ group. * *p* < 0.05, ** *p* < 0.01, *** *p* < 0.001, **** *p* < 0.0001, compared between the groups.

**Figure 2 nutrients-15-04840-f002:**
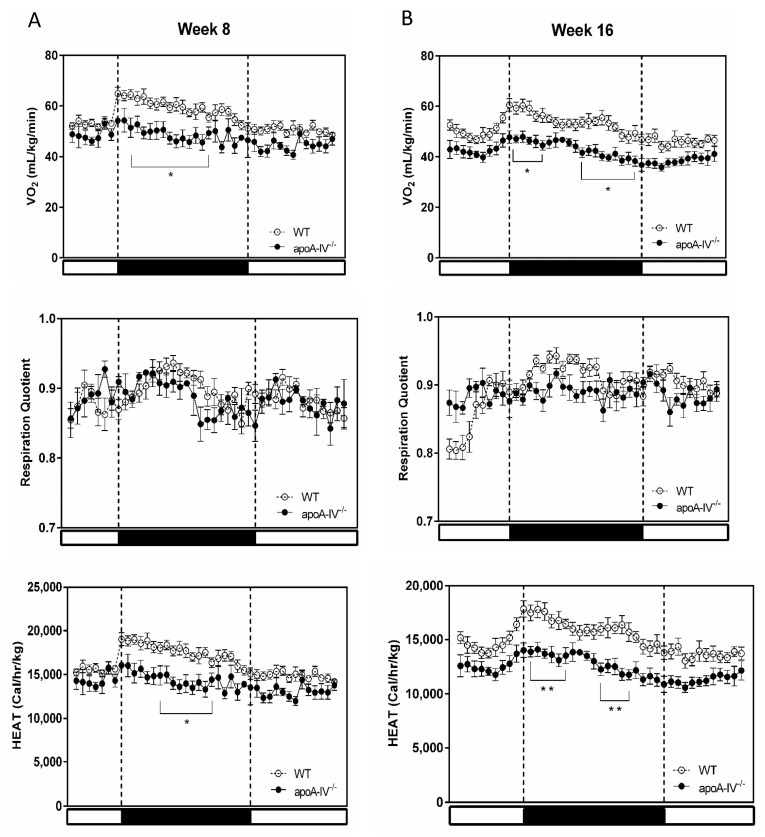
Indirect calorimetry during the period of HFD feeding. Comparison of metabolic rate between the apoA-IV^−/−^ mice and the WT mice on Weeks 8 (**A**) and 16 (**B**). Dark regions with two dash lines represent the dark period of the 24-h cycle. Values are expressed as means ± SEM. *n* = 11 in WT group, *n* = 7 in apoA-IV^−/−^ group. * *p* < 0.05, ** *p* < 0.01, compared between the groups.

**Figure 3 nutrients-15-04840-f003:**
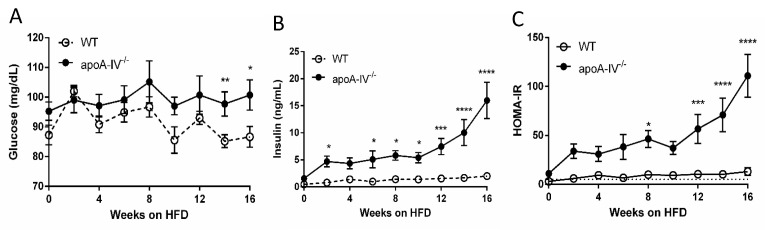
Glucose homeostasis during 16-week HFD. Comparison of plasma glucose level (**A**), insulin level (**B**), and HOMA-IR (**C**) between the apoA-IV^−/−^ and the WT mice. Values are expressed as means ± SEM. *n* = 11 in WT group, *n* = 7 in apoA-IV^−/−^ group. * *p* < 0.05, ** *p* < 0.01, *** *p* < 0.001, **** *p* < 0.0001, compared between the groups.

**Figure 4 nutrients-15-04840-f004:**
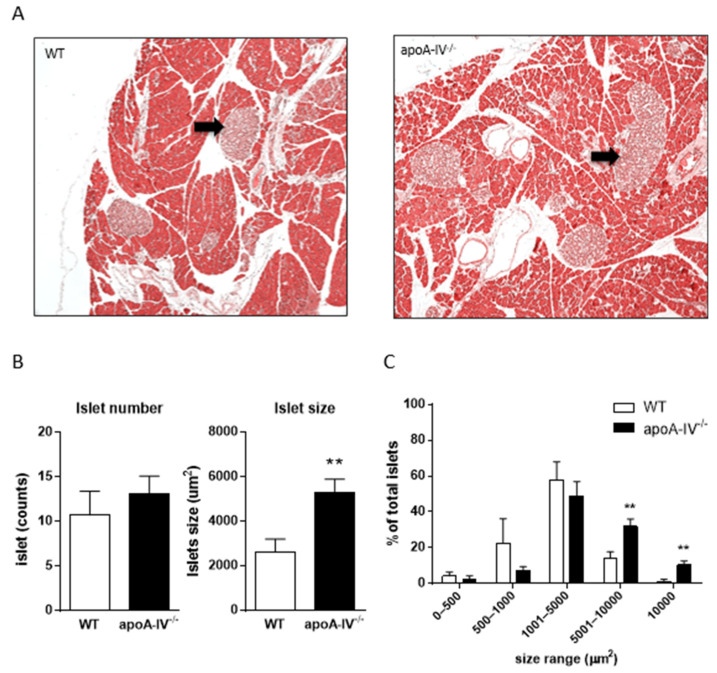
Morphology of pancreatic islets at the end of 16-week HFD feeding. (**A**) The islet sections with H&E staining from the apoA-IV^−/−^ and WT mice. Black arrow points to the islet. Comparison of islet number, size difference (**B**), and total islet size range (**C**) between the groups. Values are expressed as means ± SEM. *n* = 7 per group. ** *p* < 0.01, compared between the groups.

**Figure 5 nutrients-15-04840-f005:**
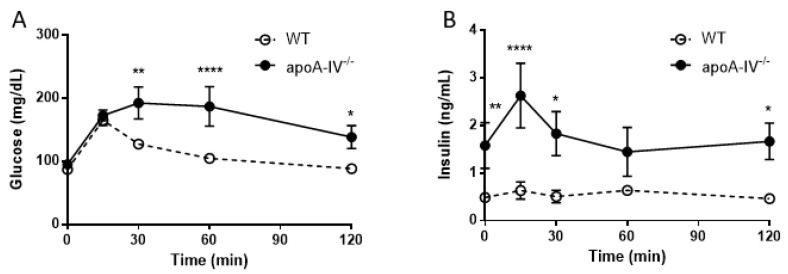
Glucose tolerance test in mice fed with chow diet. Comparison of glucose level (**A**) and insulin level (**B**) between the apoA-IV^−/−^ mice and the WT mice. Values are expressed as means ± SEM. *n* = 11 in WT group, *n* = 7 in apoA-IV^−/−^ group. * *p* < 0.05, ** *p* < 0.01, **** *p* < 0.0001, compared between the groups.

**Figure 6 nutrients-15-04840-f006:**
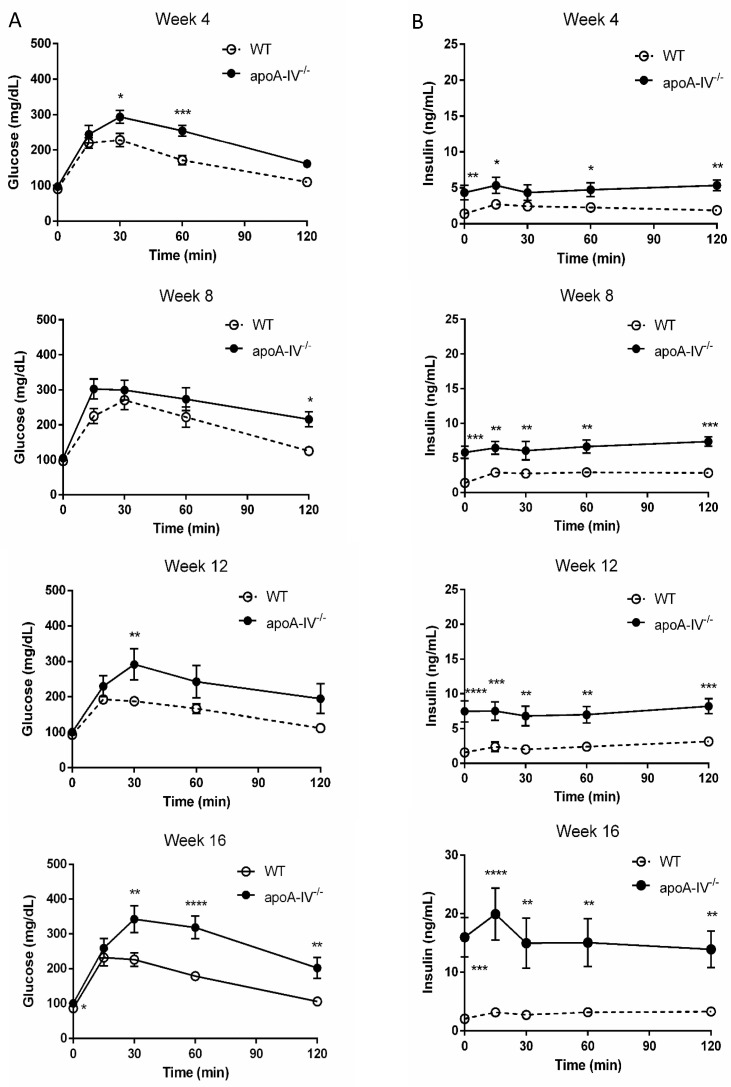
Glucose tolerance test during 16-week HFD. Comparison of glucose level (**A**) and insulin level (**B**) between the apoA-IV^−/−^ mice and the WT mice on weeks 4, 8, 12, and 16. Values are expressed as means ± SEM. *n* = 11 in WT group, *n* = 7 in apoA-IV^−/−^ group. * *p* < 0.05, ** *p* < 0.01, *** *p* < 0.001, **** *p* < 0.0001, compared between the groups.

**Figure 7 nutrients-15-04840-f007:**
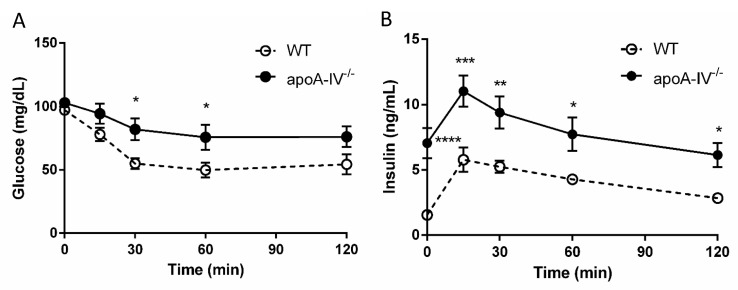
Insulin sensitivity test in mice on Week 13 of HFD. Comparison of glucose levels (**A**) and insulin levels (**B**) between the apoA-IV^−/−^ mice and the WT mice on Week 13. Values are expressed as means ± SEM. *n* = 11 in WT group, *n* = 7 in apoA-IV^−/−^ group. * *p* < 0.05, ** *p* < 0.01, *** *p* < 0.001, **** *p* < 0.0001, compared between the groups.

**Figure 8 nutrients-15-04840-f008:**
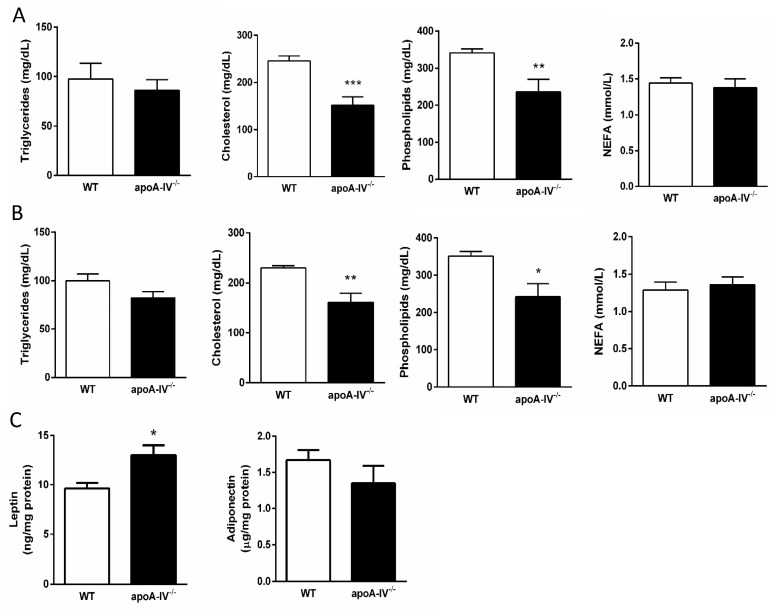
Circulating lipid profile during 16-week HFD. Comparison of triglyceride, cholesterol, phospholipid, and NEFA levels between the apoA-IV^−/−^ mice and the WT mice on Weeks 8 (**A**) and 16 (**B**). Comparison of leptin and adiponectin levels on Week 16 (**C**). Values are expressed as means ± SEM. *n* = 7 per group. * *p* < 0.05, ** *p* < 0.01, *** *p* < 0.001, compared between the groups.

## Data Availability

Data are contained within the article and Appendix A.

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
