# Peer review of "Apolipoprotein A-IV-Deficient Mice in 129/SvJ Background Are Susceptible to Obesity and Glucose Intolerance"

_nutrients, 2023, doi:10.3390/nu15224840_

Round 1

Reviewer 1 Report

Comments and Suggestions for Authors

The study involved mice lacking apoA-IV within a specific genetic background (129/SvJ). These mice exhibited weight gain, glucose metabolism issues, and insulin resistance, distinguishing them from similar mice in a different genetic context (C57BL/6J). It's worth noting that the roles of apoA-IV in regulating energy balance and glucose metabolism have been previously explored by other research groups. While this manuscript does introduce some novel findings, it may not meet the criteria for acceptance in this journal. Additionally, there appear to be certain shortcomings in the research design.

To gain a better understanding of the roles of apoA-IV in energy balance and glucose metabolism, I recommend referring to previously published papers on the subject.

1)    Apolipoprotein A-IV improves glucose homeostasis by enhancing insulin secretion (109 (24) 9641-9646, https://doi.org/10.1073/pnas.1201433109)

2)    Reduced Diet-induced Thermogenesis in Apolipoprotein A-IV Deficient Mice (Int J Mol Sci. 2019 Jun 28;20(13):3176. doi: 10.3390/ijms20133176)

3)    Apolipoprotein A-IV involves in glucose and lipid metabolism of rat (https://doi.org/10.1186/s12986-019-0367-2)

4)    Apolipoprotein A-IV reduced metabolic inflammation in white adipose tissue by inhibiting IKK and JNK signaling in adipocytes (Mol Cell Endocrinol. 2023 Jan 1:559:111813. doi: 10.1016/j.mce.2022.111813)

Other comments.

1.          To make accurate comparisons, it's essential to provide data not only on the percentages of fat and lean components in body composition but also their respective weights in grams (g). Clear quantitative data are necessary for a thorough analysis of body composition.

2.          The absence of a control group of mice on a standard chow diet in these experiments makes it challenging to confirm whether the high-fat diet-induced obesity and metabolic disorders. Including a control group is vital for a clear interpretation of the results.

3.          When starting the high-fat diet (HFD), ensure that there are no noticeable differences in body weight among the mice. This is crucial to avoid confounding factors related to initial weight variations.

4.          Since the sleep and activity patterns of mice can influence fasting conditions, please explicitly mention the timing of fasting in the figure legend for clarity.

Comments on the Quality of English Language

It looks OK

Reviewer 2 Report

Comments and Suggestions for Authors

Wang et al present a manuscript of high conceptual importance.

Earlier work from the group investigated the metabolic status of Apolipoprotein A-IV (apoA-IV) KO mice in the typically used C57BL/6J (Black 6) strain.

Therein, authors demonstrated improved glucose homeostasis upon apoA-IV deletion.

Now, after extensive backcrossing of the deleted allele into the 129/SvJ, authors report on the metabolic profile of apoA-IV deletion, but in a different genetic background.

While apoA-IV deletion in the black 6 had a somehow favorable effect on metabolism, apoA-IV deletion in 129/SvJ promotes obesity and leads to glucose intolerance, as demonstrated here through a comprehensive metabolic characterization.

Overall, in this manuscript, there are no really experimental points that should be addressed, or other issues related to the quality of date presentation.

Yet, a commentary that I would like to propose is the following: in the conclusion remarks (from line 379), authors write: “….. highlight the impact of genetic background……findings will, therefore, stimulate future research in individualized treatments of obesity and diabetes in humans in the era of precision medicine.”

If consider from another perspective, the present study explores the effects of a gene deletion on yet another inbred mouse strain, and I suggest that a mention on this point should be made.

Ideally, to better represent human genetic variability, gene deletion studies should be done on outbred strains.

Comments on the Quality of English Language

Just check minor spelling mistakes. Line 100: "Proctor and Gamble".... "Procter " ?

Reviewer 3 Report

Comments and Suggestions for Authors

The manuscript, titled "Apolipoprotein A-IV-deficient mice in 129/SvJ background are susceptible to obesity and glucose intolerance," provides a thought-provoking exploration of apolipoprotein A-IV (apoA-IV) and its role in metabolic regulation, with a specific emphasis on how the genetic background influences observed effects. The study's findings are indeed compelling, shedding light on the significance of apoA-IV in metabolic processes. The observation that 129/SvJ apoA-IV-/- mice exhibited heightened weight gain and delayed glucose clearance, even when on a standard chow diet, is particularly noteworthy. This finding underscores the substantial impact of apoA-IV deficiency on parameters such as obesity, increased food intake, reduced energy expenditure, and physical activity during a high-fat diet study. Furthermore, the discovery of insulin resistance and severe glucose intolerance in apoA-IV-/- mice, despite elevated insulin levels, is a crucial result that merits further investigation.

However, it would enhance the completeness of the study and provide a more comprehensive understanding of the findings if the following experiments or data were included:

1.       Measurement of mRNA and protein levels of key genes that regulate lipid metabolism is advisable. This additional data would contribute to a more comprehensive picture of the molecular mechanisms underlying the observed metabolic effects in apoA-IV-/- mice.

2.       The inclusion of RNA sequencing data would be highly beneficial. It could offer insights into the mechanisms through which apoA-IV regulates food intake, especially in 129/SvJ mice compared to C57BL/6J mice. This added information would enrich the discussion of the genetic background's influence on apoA-IV's metabolic effects.

Round 2

Reviewer 1 Report

Comments and Suggestions for Authors

The authors respond to the comment properly although some issues still exist. Nevertheless, the study deserves to be published in this journal.

Comments on the Quality of English Language

It looks OK